# Merging Ligand-Based and Structure-Based Methods in Drug Discovery: An Overview of Combined Virtual Screening Approaches

**DOI:** 10.3390/molecules25204723

**Published:** 2020-10-15

**Authors:** Javier Vázquez, Manel López, Enric Gibert, Enric Herrero, F. Javier Luque

**Affiliations:** 1Pharmacelera, Plaça Pau Vila, 1, Sector C 2a, Edificio Palau de Mar, 08039 Barcelona, Spain; enric.gibert@pharmacelera.com; 2Department of Nutrition, Food Science and Gastronomy, Faculty of Pharmacy and Food Sciences, Institute of Biomedicine (IBUB), and Institute of Theoretical and Computational Chemistry (IQTC-UB), University of Barcelona, Av. Prat de la Riba 171, E-08921 Santa Coloma de Gramanet, Spain; 3AB Science, Parc Scientifique de Luminy, Zone Luminy Enterprise, Case 922, 163 Av. de Luminy, 13288 Marseille, France; manel.lopez@ab-science.com

**Keywords:** ligand-based techniques, structure-based methods, combined strategies, virtual screening, drug discovery

## Abstract

Virtual screening (VS) is an outstanding cornerstone in the drug discovery pipeline. A variety of computational approaches, which are generally classified as ligand-based (LB) and structure-based (SB) techniques, exploit key structural and physicochemical properties of ligands and targets to enable the screening of virtual libraries in the search of active compounds. Though LB and SB methods have found widespread application in the discovery of novel drug-like candidates, their complementary natures have stimulated continued efforts toward the development of hybrid strategies that combine LB and SB techniques, integrating them in a holistic computational framework that exploits the available information of both ligand and target to enhance the success of drug discovery projects. In this review, we analyze the main strategies and concepts that have emerged in the last years for defining hybrid LB + SB computational schemes in VS studies. Particularly, attention is focused on the combination of molecular similarity and docking, illustrating them with selected applications taken from the literature.

## 1. Introduction

Predicting with chemical accuracy the biological activity that a small drug-like compound can attain against its target is a major challenge in drug discovery. In the late stages of lead optimization, this task can be accomplished by resorting to enhanced sampling techniques, such as free energy calculations [1,2,3], which can estimate the binding affinity between a ligand and its macromolecular target. Remarkably, the effort spent in developing robust algorithms in conjunction with efficient configurational sampling methods permit to estimate the binding affinity with a chemical accuracy close to the 1 kcal/mol limit [4,5,6,7], though at the expense of a significant computational cost that prevents their application in large datasets. Nevertheless, attempts have been made to alleviate this limitation through the development of automated workflows for the in silico prediction of binding affinities [8,9], which will facilitate the usage of these sophisticated techniques to nonexpert researchers in computational chemistry.

A different scenario occurs in the early stages of drug discovery, where attention is focused on the identification of potential hit compounds endowed with a promising activity against a druggable target or, alternatively, the search of novel chemotypes that may lead to innovative strategies in the treatment of diseases and pathological disorders. In these stages, the suitability of several computationally demanding methods, such as steered molecular dynamics (MD) and quantum mechanical-based approaches, has been explored, generally when the interest is focused on a reduced set of compounds [10,11,12]. Nevertheless, when one keeps in mind the diversity of the chemical universe that can a priori be explored [13,14,15], the ability to discriminate between actives and inactives still represents a formidable challenge that makes it necessary to resort to simplified computational approaches. At this point, it is worth noting that the number of different compounds that could be synthesized has been estimated to be around 10^20^–10^24^ molecules [16]. The vast amount of drug-like chemical libraries can be explored using in silico virtual screening (VS) techniques, which encompass a variety of computational algorithms and formalisms in the search of novel bioactive molecules. They have proven their applicability in numerous studies, leading to hit rates competitive with the results derived from experimental high-throughput screening and at a much lower cost [17,18,19].

VS techniques can be grouped into two major categories, depending on the available structural information. The term structure-based virtual screening (SBVS), often denoted as target-based VS, encompasses methods that exploit the three-dimensional (3D) structure of the target. The most widely used SBVS technique is molecular docking, which uses the structural and chemical complementarity resulting from the interaction between a fragment-like or drug-like compound and its target receptor, predicting the preferred pose of ligands in the binding site through the use of scoring functions, often supplemented with pharmacophoric constraints [20,21,22,23]. On the other hand, ligand-based virtual screening (LBVS) relies on the structural information and physicochemical properties of the chemical scaffold of known active and inactive molecules, which are examined under the molecular similarity principle [24]. Accordingly, the relationships between compounds in a given library and one or more known actives are examined by similarity measurements using suitable molecular descriptors. These measurements can be performed based on 1D and 2D descriptors, generally encoding information about the chemical nature of compounds and their topological features [25,26,27], and 3D descriptors associated to molecular fields [28,29,30,31], shape and volume [32,33], and pharmacophores [28,34].

The combined integration of SBVS and LBVS techniques may be a promising strategy when data about both the structure of ligand-target complexes and similarity relationships to active compounds are available, leading to a holistic framework suitable to enhance the success of drug discovery projects [35,36]. As an example of the potential impact of combining SBVS and LBVS, we limit ourselves to cite a couple of representative studies. The first is the work by Spadaro et al. [37], who used a pharmacophoric model derived from the analysis of X-ray crystallographic data in conjunction with LBVS techniques for disclosing novel inhibitors of the 17β-hydroxysteroid dehydrogenase type 1 (17β-HSD1) enzyme, leading to the identification of a keto-derivative compound with an inhibitory potency in the nanomolar range (Figure 1A). In the second example, Debnath et al. [38] used a combined VS strategy to identify selective non-hydroxamate histone deacetylase 8 (HDAC8) inhibitors (Figure 1B). To this end, a database of 4.3 × 10^6^ molecules was explored using a pharmacophore model, and the top 500 hits retrieved were filtered using ADMET (Absorption, Distribution, Metabolism, Excretion and Toxicity) criteria. The selected compounds were subsequently assessed by molecular docking. Among the final hits selected for in vitro biological evaluation, compounds SD-01 and SD-02 inhibited the HDAC8 enzyme with IC_50_ (i.e., the concentration of inhibitor that gives half-maximal response) values of 9.0 and 2.7 nM, respectively. These two examples suffice to demonstrate that a judicious choice of LB and SB techniques, adapted to the available information about the ligands and target, may be powerful in disclosing drug-like compounds.

Several strategies have been proposed to combine LBVS and SBVS in order to reinforce the mutual complementarity of these approaches and palliate their individual weaknesses [39,40,41]. The major shortcoming in LBVS is the bias toward the reference template, which may result in overfitting to the input structures. When a pharmacophore is used to guide the screening of the compounds, the chemical features of the ligands present in the training set may affect the optimal choice of the pharmacophoric restraints. Moreover, the available activity data may turn out to be inadequate for selecting a structural and functional pool of compounds, often limited by the absence of data relative to poorly active or inactive compounds, which may be valuable to calibrate the merits of the pharmacophore model in distinguishing between actives and inactives. On the other hand, accounting for protein flexibility is a major drawback for docking methods. The binding site of a protein is flexible and can adopt diverse conformational states, generally at the level of side chain residues but often also involving structural changes in loops and the remodeling of secondary structural elements induced upon ligand binding [42,43,44,45]. Furthermore, the outcome of docking studies may be largely affected by the identification of water molecules that mediate the interactions of the ligand in the binding pocket, making it necessary to explore the potential role of bridging waters or networks of ordered waters in docking calculations [46,47,48,49,50]. On the other hand, providing an accurate score and even estimating the binding affinity at a reasonable cost compatible with the screening of large chemical libraries is still challenging for docking methods [51,52,53,54]. Finally, the outcome of LBVS and SBVS also appear to exhibit a strong target dependency [55,56]. For the sake of brevity, a detailed discussion of these weaknesses is omitted here, and the reader is addressed to previous studies in the literature [57,58,59,60].

In this context, searching for computational strategies that can mitigate the limitations of LB and SB methods has been actively pursued in the last years. One alternative is to focus the screening effort on targeted chemical libraries that would facilitate the task of hit identification [61,62,63,64,65]. This can be achieved via automated algorithms of molecular generation or de novo design, often assisted by artificial intelligence techniques, which aim to create sets of compounds endowed with properties similar to the structural and chemical features found in real cases, including a bias toward specific ranges of physicochemical properties or toward compounds active against a given target. Alternatively, a balanced combination of LB and SB methods may be devised to exploit synergistically the merits of these VS techniques, while counterbalancing their limitations, in order to increase the success rate in the screening of large chemical libraries.

Here, our attention is focused on the methodologies and computational approaches undertaken to enrich the outcome of VS by combining LB and SB techniques. In particular, we review the main strategies that have been proposed combining molecular similarity and docking. The strengths and weaknesses of the combined approaches are illustrated by selecting representative studies reported in the literature, primarily dealing with the efforts reported in the last five years. Overall, the aim of this review is to provide useful guidelines for the application of combined LB and SB methods in drug discovery.

## 2. LB and SB Strategies in VS

Different schemes can be adopted to combine LB and SB methods. The classification proposed by Drwal and Griffith will be adopted in this review [40]. Accordingly, the discussion of the combined LB and SB strategies can be completed following three main categories: sequential, parallel, and hybrid, which are summarized in Figure 2.

(i) Sequential approaches divide the VS pipeline in consecutive steps with the aim to perform a progressive filtering in the library of chemical compounds toward the most promising candidates, which will be selected for biological testing at the end of this multi-step process. Generally, prefiltering is performed at the beginning of the VS process using LB techniques due to their reduced computational cost, whereas the most computationally demanding SB methods are exploited in the final stages of the selection process. Thus, this strategy attempts to optimize the tradeoff between the computational expensiveness and the complexity of the formalism that underlies the filtering technique along the VS process. However, they do not exploit all the available information at once and maintain the limitations of the individual methods.

(ii) In the parallel approach, both LB and SB methods are run independently, and the best candidates identified from each separate method are selected for biological testing. Swann et al. reported a prospective application of this approach in 2011 [66], and subsequent studies have examined distinct functional forms for combining the ranks obtained from LB and SB methods (see below). In particular, the compounds obtained in the final rank order lead to meaningful increases in both performance and robustness over the single-modality approaches, but the results also demonstrate the sensitivity of the performance to the target structural details (i.e., the nature of the template ligand in measurements of molecular similarity and the reference protein pocket in docking studies) [67,68].

(iii) Finally, the hybrid strategies comprise approaches that represent a true combination of LB and SB techniques into a standalone method. Two main combinations have been followed to achieve this goal: (i) interaction-based methods and (ii) similarity-docking methods. The former translates the observed protein-ligand interactions into pharmacophoric features and quantitative structure-activity relationship (QSAR) models [39,69,70], which have been used for several applications, such as VS, the profiling of ligands, the analysis of pseudo-receptors, and de novo designs [71,72,73,74]. On the other hand, the combination of molecular similarity and docking techniques has been examined in the last years as an alternative procedure to assess the reliability of predicted poses of ligands by measuring the overlay against suitable templates [75,76,77,78,79,80].

## 3. Sequential LB and SB Methods

High-throughput VS may be computationally demanding when large sets of compounds have to be evaluated. In this scenario, decomposing the VS pipeline into a multi-step process can be valuable to reduce progressively the number of compounds and enrich the chemical library toward the most promising scaffolds before screening with more expensive methods.

LB techniques are generally used in the prefiltering step, as illustrated in different works that have exploited 2D fingerprints [81,82], 3D molecular similarity [83,84,85] and pharmacophore models [86,87,88]. To enhance the drug-likeness of the compounds, knowledge-based in silico ADMET or pan-assay interference compounds (PAINS; [89]) filters can also be applied. For cases with a reduced number of compounds, the results obtained from the SBVS can be further refined, resorting to the structural stability observed in MD simulations [90,91,92,93].

As an example that illustrates the sequential application of LB and SB techniques, Khan et al. [84] performed multi-step LBVS and SBVS to identify G protein-coupled estrogen receptor-1 (GPER-1) modulators (Figure 3A). LBVS was performed based on a GPER-1 selective agonist (1-((3a*R*,4*S*,9b*S*)-4-(6-bromobenzo[*d*][1,3]dioxol-5-yl)-3a,4,5,9b-tetrahydro-3*H*-cyclopenta(c)quinolin-8-yl)ethan-1-one) as a query model for screening of the eMolecules library (about 7.2 million compounds used in [84]) using Rapid Overlay of Chemical Structures (ROCS; [32]) and electrostatic potential screening (EON; [94]). Then, after generation of a GPER-1 homology model, FRED [95,96] was used to screen the top-scored hits from LBVS. Next, the top-ranked hits retrieved by molecular docking were clustered based on the similarity between their scaffolds. Finally, the prospective validation in SK-BR-3 and MCF-7 cell lines resulted in two compounds with an EC_50_ (i.e., effective drug concentration that gives half-maximal response) antiproliferative activity in the micromolar range.

Alternative LB and SB sequential protocols have also been adopted, as in the study by Dawood et al. [97], where the SBVS was followed by a LBVS (Figure 3B). An in-house database of 1720 phytochemicals used in traditional Egyptian medicine was screened to search for inhibitors of the human aromatase enzyme. The initial size of the library allowed the direct use of molecular docking using Glide [98,99,100]. Subsequently, a LB pharmacophore was used to filter the ranked compounds with PHASE [101,102]. In vitro testing revealed that the methylene chloride extract of *Artemisia annua* showed the most significant aromatase inhibitory activity with an IC_50_ of 2.2 μg/mL, thus opening a path for the use of secondary metabolites in the search for new therapeutic leads. 

## 4. Parallel LB and SB Approaches

The application of different LB and SB methods generates distinct sets of ranked compounds for the same target. Given that there is no single method that consistently ranks a database of compounds in the best decreasing order, a combination of the ranks obtained from multiple LB and SB searches into a single ranking could lead to a better overall enrichment and a wider diversity of hit structures [103].

In this context, LBVS approaches have been combined under the framework of data fusion [104,105], targeting the search for new entities, drug repurposing, polypharmacology, and safety profile analysis [106,107,108,109]. With regard to SBVS, distinct methods have been examined to yield a “consensus scoring” [110,111,112,113], relying on three main strategies: (i) the same docked poses have been evaluated with different scoring functions to build the final ranking, (ii) the results obtained for an ensemble of different protein structures of the same target have been combined to obtain a final score, and (iii) multiple docking methods have been used against a single-protein structure [54,114,115].

Efforts have also addressed the development of parallel protocols for combining LB and SB methods [66,67,103]. Table 1 summarizes different fusion strategies that have been adopted to combine the results of LB and SB techniques in benchmarking studies. The parallel selection seems to perform better than other rank fusion metrics, although the quality of the structural information and the specific physicochemical features of the target system may influence the overall performance. For instance, Tan et al. [116] evaluated the performance of a parallel protocol that combined docking and similarity search calculations using 2D fingerprints on nine target enzymes. The results were combined through rank fusion, where the ranks from docking and similarity searching were added to generate the final ranking, or the parallel selection method, where compounds are alternately selected according to the ranks obtained separately for LB and SB screenings. These combinations yielded an overall improvement in compound recall in 25% of the calculations. Furthermore, parallel selection was found to be more effective than rank fusion.

Swann et al. examined the combination of LBVS (2D graph-based extended connectivity fingerprint (ECFP6) [117] and ROCS) and SBVS (chemical Gaussian overlay, CGO [95]) within a probabilistic framework that returns a quantitative likelihood (or probability) of observing bioactivity for the selected compounds [66]. The analysis of the results obtained for a set of 18 targets showed that the retrieval rates for the cumulative probability (obtained from the fusion of the individual LB and SB values) are equal to or better than the highest retrieval rate achieved with any single method. Similar trends were observed for an additional external validation set of six targets, and, importantly, the method was successful in the identification of novel hit compounds in a prospective study performed against four targets not included in the training and validation sets.

A number of studies dealing with the application of the parallel strategy in the search of novel hits have been reported in the last years [118,119,120,121]. An illustrative example is the work by Vucicevic et al. [119], who reported the identification of compounds with anticancer potential effects through a parallel LB and SB screening protocol (Figure 4A). Starting from a large virtual library with more than 9 × 10^6^ compounds, those molecules that showed good ranking in both approaches were selected for biological testing. The most active compound exhibited a cytotoxic profile similar to the positive control and enhanced the apoptotic response to doxorubicin, thus representing an adjuvant chemotherapeutic strategy for doxorubicin-insensitive cancers.

Finally, a more recent example is the work by Costa et al. [121], where they performed a parallel VS application followed by MD simulations in the search of a novel compound able to inhibit human immunodeficiency virus type 1 (HIV-1) reverse transcriptase (RT) RNA-dependent DNA polymerase activity (Figure 4B). More than 143,000 natural compounds commercially available in the ZINC database were screened. As a result, 20 hit molecules were chosen and tested in biochemical assays. However, instead of merging the output rankings, compounds shared in both LB and SB VSs were selected. Among them, three compounds were identified as novel non-nucleoside RT inhibitors in the low micromolar range.

As a final remark, let us note that, compared to sequential methods, parallel LB and SB screenings imply a larger computational cost, as several VS techniques have to be simultaneously run in order to derive their respective rankings, which will subsequently be used to generate the final selection.

## 5. Hybrid Approaches

As noted above, hybrid LB and SB strategies can be grouped into two major categories, which are denoted as (i) interaction-based approaches and (ii) similarity-docking methods.

### 5.1. Interaction-Based Methods

These methods rely on the identification of patterns of protein–ligand interactions, which are subsequently used in the screening of compounds through the usage of pseudo-receptor and pseudoquery methods. Since SB information is not effectively incorporated in pseudo-receptor models, we limit ourselves to giving a brief description for the sake of completeness but omit a detailed discussion, which can be found elsewhere [73].

Pseudo-receptor methods rely on the mapping of the potential interactions that may be formed by a set of reference ligands suitably aligned in their bioactive conformation to mimic their overlaid arrangement in the binding pocket [122,123,124]. This process leads to a rough definition of the overall shape and key anchoring points of the binding pocket, which can be exploited for the screening of chemical libraries. The performance of these models is strongly affected by the chemical space of the ligand dataset and the overlay of the ligands. The model can only account for those features present in the starting set of ligands, and the superposition of ligands is sensitive to minor modifications in the chemical scaffold, especially for highly flexible ligands.

In contrast with the preceding approaches, pseudoquery methods exploit the experimental structures of protein–ligand complexes in order to extract a profile of the interaction pattern established by the ligands bound to the protein target. This pattern is generally translated into fingerprints that encode ligand–target interactions or, alternatively, into pharmacophoric features and then used in similarity searches to find ligands that match the interaction pattern [123,124,125,126,127,128,129,130,131] (see Table 2 for a brief description of several formalisms). In addition, the search of novel hits can be performed, imposing constraints related to the shape and volume of the binding site. 

The pioneering methods included key elements of the protein–ligand complex, such as the formation of hydrogen bonds, hydrophobic or aromatic interactions, or contacts with acidic and basic groups, often supplemented by isocontours of the binding site. As an example, Salentin et al. [138] resorted to PLIP to perform a pharmacophoric search of over more than 170,000 complexes using protein-ligand interaction profiles, leading to the disclosure of the FDA-approved malaria drug amodiaquine as the top-ranking hit, which was subsequently validated as a potential anticancer agent showing inhibitory activity on the target protein Hsp27. This demonstrates the potential of pseudoquery methods for drug repurposing.

Recent methods have evolved to include solvation and entropy effects. For instance, Tran-Nguyen et al. [131] included in their pseudoquery pharmacophoric tool the desolvation component of the protein–ligand interaction energy using a Poisson−Boltzmann treatment. Furthermore, the analysis was decomposed in three consecutive steps: (i) the detection of druggable cavities at the surface of the protein target and the identification of pharmacophoric features, (ii) the generation of cavity-based pharmacophore queries in the 3D space, and (iii) molecular alignment exploiting the cavity-based feature. The proposed pharmacophoric model was benchmarked using DUD-E [139]. Unique chemotypes were retrieved from high-throughput VS, being as efficient as state-of-the-art docking [140] and shape-matching [32] methods in both pose prediction and ranking power.

As noted above, pseudoquery methods have also exploited interaction fingerprint patterns (IFP) containing information about the contacts of the ligand with the protein, thus condensing the 3D structural binding information into a 1D binary string, leading to a drastic reduction in the cost of VS. This is exemplified by the Structural Interaction Fingerprint (SIFt; [132]) method, where crystallographic ligands are divided into two groups of fragments: (i) atoms involved in protein–ligand interactions (interaction fragments, IFs) and (ii) fragments generated by the random deletion of ligand atoms used as a control. Then, for each ligand and the corresponding fragments, MACCS (Molecular ACCess System) structural keys [141] were calculated and used as a fingerprint for similarity searching. The results of their validation work suggested that IFs used as templates can increase the similarity search performance of conventional structural fingerprints. SIFt is included in Arpeggio [142], a web server for the analysis of protein interactions with small-molecule ligands, proteins, and DNA. 

The concept of IF has been adopted in alternatives models to outperform conventional scoring functions in predicting the correct poses for drug-like compounds [143], similarity-based screening [137,144,145,146,147,148], binding/unbinding kinetics [146], and drug resistance [147].

### 5.2. Similarity-Docking Strategies

An important challenge in VS is to create accurate scoring and ranking functions to identify hit compounds active against specific targets. In this context, similarity measurements between compounds can be used to assist molecular docking to score sampled poses [148,149,150,151,152,153,154] and to discriminate between active and inactive molecules [155,156]. Exploiting the synergy between molecular similarity and docking has received increasing interest in the last years, leading to hybridized tools, such as HomDock [157] and Hybrid [96].

#### 5.2.1. Predicting the Pose of Ligands

A direct application of merging molecular similarity and docking is related to improving the prediction of the ligand pose in the binding pocket. At this point, exploiting the experimental information on the binding mode of active compounds has been shown to enhance the performance of predicting the pose of drug-like compounds [158,159,160]. For instance, the participants of the Drug Design Data Resource (D3R) Grand Challenge 3 were challenged in predicting the binding poses of 24 cathepsin S ligands. Kumar and Zhang [161] tested the performance of three methods (PoPSS [150], CDVS [162], and PoPSS-Lite) based on the concept of ligand 3D shape similarity. PoPSS evaluates the shape similarity with existing crystallographic compounds bound to the target protein for predicting the poses of query ligands with unknown binding modes. The ligand with the highest shape similarity score was selected and placed into the binding pocket. After ligand placement, side-chain residues of the binding pocket were repacked based on the query ligand conformation, followed by Monte Carlo energy minimization of the protein-ligand complex. Finally, ligand-bound structures were scored using the Rosetta energy function [163]. For CDVS and PoPSS-Lite, 3D shape similarity calculations were also used to identify the ligand pose. However, once the suitable ligand–receptor pair was identified, CDVS performed a standard docking using Glide, and PoPSS-Lite refined the pose with an energy minimization. PoPSS-Lite exhibited an excellent performance in this challenge, leading to the lowest mean root mean square deviation (RMSD) values between the native and predicted poses. Moreover, CDVS and PoPSS were located among the best 15 methods tested for both metrics.

Shape similarity between the ligand conformation and the crystallographic ligand is the most common scheme adopted for guiding the pose prediction. However, other similarity measurements and methodological refinements have been explored. As an example Jacquemard et al. defined a benchmark constituted by 2376 high-quality structures representing 64 proteins and compared the performance of three rescoring schemes applying the similarity of IFP, graph matching of interaction patterns (GRIM [137]), and ROCS [78] (Figure 5). GRIM and ROCS were more efficient than IFP rescoring based on 2D fingerprints, even when the comparison involved structurally dissimilar molecules. In addition, the speed of calculation for all the methods was improved, facilitating the processing of a large number of poses.

The search for methodological innovations is also exemplified by Kumar and Zhang [151], as they modified PoPSS to account for water-mediated protein–ligand interactions using a continuum Poisson-Boltzmann (PB) solvation model, leading to the PoPSS-PB model. PoPSS-PB demonstrated an excellent performance in D3R GC4, with mean and median RMSDs of 1.20 (ranked 10th out of 74) and 1.13 (ranked 9th out of 74) Å, improving the performance obtained for PoPSS and PoPSS-Lite.

Finally, Varela-Rial et al. [164] also evaluated in the D3R Grand Challenge 4 an algorithm named SkeleDock to define the binding mode based on the structure of a protein−ligand complex. The algorithm defines graphs for the query and the template molecules, and then, these graphs are compared to extract a common subgraph, which describes a continuous set of atoms whose element (node) and bonds (edges) are equivalent in the two molecules (Figure 6). Thus, a mapping that links atoms in query and template compounds can be identified, facilitating the conformational adjustment of the atoms in the query ligand onto those in the template molecule, whereas atoms in the query molecule with no equivalent counterpart in the template are positioned by using a tethered template docking protocol. The algorithm was ranked 15th out of 74 according to the mean RMSD (1.33 Å) and 9th according to the median RMSD (1.02 Å).

#### 5.2.2. Similarity-Guided Score Scheme

In addition, to assist the prediction of the ligand pose in the binding cavity, similarity measurements can also be used as a weighting factor in the reranking of the docked compounds. In fact, considering that LB 3D-shape matching algorithms often produced better enrichments than docking, assessing the overlay of docked poses relative to known crystallographic ligands could be valuable to retrieve active compounds in screening studies. To this end, the scoring function in docking calculations could be supplemented with 3D molecular similarity measurements to build the final ranking in the VS process, taking into account that positive candidates accommodated in the active site are expected to share similar structural and physicochemical features that resemble those of known actives in cocrystal structures and improve the ranking of the screened ligands.

The first implementations of this LB and SB scheme in docking programs were performed by Marialke et al. [157] and McGann [96] in the development of HomDock and Hybrid, respectively. HomDock is a combination of optimization methods with graph-based molecular alignment (GMA; [165]) that superposes a query molecule on a rigid template. GMA places candidate ligands over the template and optimizes their placement in the field of the protein. Then, the ligands are ranked according to their interaction with the protein and/or their structural similarity with the ligand. On the other hand, Hybrid uses an exhaustive search algorithm, treating ligand and protein structures as rigid bodies. Both the protein and ligand flexibility are addressed through multiple conformers. Subsequently, the CGO ligand-based scoring function is applied. CGO scores based on how well the docked molecule matches the shape and 3D arrangement of the chemical features of the crystallographic ligand bound to the active site.

In another vein, Anighoro and Bajorath published a series of comparative studies where the best poses of commercial docking software are directly scored using 3D similarity methods, such as the whole-ligand 3D shape similarity and protein–ligand IFP similarity [76,155,156]. The protocol was validated by performing retrospective VS calculations for different targets, including dihydrofolate reductase, glucocorticoid receptor, HIV-1 protease, vascular endothelial growth factor receptor-2, adenosine A2A receptor, and β2 adrenergic receptor. As noted in Figure 7, the hybridized approach yielded better performance in retrieving active compounds against the targets included in the validation set. Thus, the results showed that ranking by whole-ligand 3D similarity calculations outperformed the force field-based ranking tested for both global performance and early enrichments. It was also shown that a ligand was less suitable as a reference for 3D similarity calculations if it contained large solvent-exposed groups not directly interacting with the target. Finally, they highlighted the importance that reference ligands should be engaged in interactions within the binding site as much as possible.

Under the same framework, our group has recently presented a 3D similarity scheme to enrich the docking performance based on the usage of lipophilic descriptors [168] determined from quantum mechanical-based continuum solvation models [80]. The 3D similarity was determined by comparing the 3D distribution of atomic lipophilicity computed by PharmScreen [31,169] (Figure 8), and the similarity measurements were exploited in conjunction with the poses obtained by using three docking programs: Glide, rDock [170], and GOLD [171]. Two hybrid algorithms were examined over 44 sets: (i) rescoring ranking (RR), where the final ranking was determined according to the score obtained from the 3D lipophilic similarity of the best pose generated by the docking method and the co-crystallized ligand, and (ii) consensus ranking (CR), where the final score of the compounds was obtained by merging the rankings provided directly from the docking method and from the RR. The results obtained support the synergy of the hybrid LB and SB approaches, as the CR consistently showed better performance than using only either the LB or SB methods. In addition, the results suggest that CR may overcome the existence of multiple binding modes differing from the experimental pose of the co-crystallized ligand.

## 6. Exploiting Chemical Libraries and Biological Data

While selecting different LB and SB strategies may provide alternative approaches to enrich the results of VS, the identification of a novel lead compound may be conditioned by the structural diversity of the chemical space encoded in compound libraries. Therefore, the choice of a representative dataset well-suited to the specific structural, physicochemical, and biological features of the macromolecular target may be crucial for the successful outcome of a VS campaign, especially keeping in mind that the available chemical libraries comprise only a small portion of the synthesizable chemical universe of compounds.

To facilitate this task, there has been continued progress in the availability of experimental data in the public domain during the last two decades [172], which is exemplified in the consolidation of curated databases of bioactive molecules with drug-like properties, as exemplified with ChEMBL [173], PubChem [174], and DrugBank [175,176], where the user may find a comprehensive compilation of diverse information, such as the chemical structure of the compounds, physicochemical properties, biological assays data, related targets, pharmacokinetics and pharmacodynamics properties, metabolic and signaling pathways, and patents. Currently, selection of the compounds can be performed through a variety of chemical databases, which can be categorized into three main groups: public, commercial (provided by vendors), and proprietary (Table 3; see references 177 and 178 for a detailed discussion).

In this context, rather than focusing the computational effort on the massive screening of larger databases, one might consider the possibility to enhance the success of LB and SB strategies in the search of novel hit compounds by resorting to the screening of targeted chemical libraries. An example is the work by Miyao et al., who reported an algorithm for the exhaustive generation of chemical structures based on inverse quantitative structure-property (QSPR)/activity (QSAR) relationships to build datasets of compounds endowed with a suitable range of desired properties [179]. The synthetic feasibility of the compounds may also be accounted from the availability of information about known chemical reactions [180,181,182]. More recently, the de novo design algorithm for exploring chemical space (DAECS) exploits the combination of a two-dimensional distribution of the chemical properties with the projection of the biological activity for a set of training compounds in order to generate structures in a specific target area of the chemical space [61,182]. A library of novel designed structures is constructed through an iterative process that involves the selection of seed structures characterized with selected chemical features and the generation of novel compounds by means of introducing slight structural changes from the seed dataset.

The design of target chemical libraries is actually an undertaking of increasing interest, as illustrated by a number of recent studies that have reported the implementation of artificial intelligence-based algorithms [63,64,65,183,184,185,186]. One of them is the development of ReLeaSE (Reinforcement Learning for Structural Evolution), which integrates a generative deep neural network with a predictive one into a joint framework for the design of novel compounds satisfying certain chemical requirements, as illustrated with the biased selection of compounds fulfilling a specific range of physical properties (i.e., melting temperature and lipophilicity) or inhibitory activity against the desired target protein (Janus protein kinase 2) [62]. Another example is the transfer-learning-based generation algorithm proposed by Amabilino et al. [186], where recurrent neural networks are used as SMILES (Simplified Molecular-Input Line-Entry System) generators and trained on a smaller set of molecules with the biological activity of interest for the design of focused libraries. On the other hand, the REINVENT code proposed by Olivecrona et al. [187] also relies on a recurrent neural network model that operates on a SMILES representation of molecules for the automated creation of molecules with predicted biological activity. Very recently, this algorithm was adapted to enable the pair-based multi-objective optimization of several molecular features based on Pareto dominance [188] and applied to the de novo design of datasets of inhibitors targeting neuraminidase, acetylcholinesterase, and the main protease of the severe acute respiratory syndrome coronavirus 2.

Another issue that deserves a brief discussion concerns the discovery of ligands able to modulate protein–protein interactions (PPIs) in the early stages of drug discovery. Given the estimated 650,000 PPIs that comprise the human interactome, the stabilization and inhibition of PPIs may represent a valuable strategy to alter the oligomerization equilibria of supramolecular protein complexes, thus altering their physiological functions in the cell [189,190,191], which may thus be exploited in the search of novel therapeutic approaches [192,193]. Nevertheless, the success of these studies may be affected by the availability of chemical libraries with ligands suitable to interact with druggable pockets at the interface of protein–protein complexes. In this context, it is worth noting the efforts made toward the design of specific databases enriched in protein–protein modulators, such as PPI-HitProfiler, which was developed to provide for any drug-like compound collection a focused chemical library enriched in putative PPI inhibitors [194], 2P2I_HUNTER_, which is a learning machine tool for filtering potential PPI modulators [195], and Fr-PPIChem, which reflects the collective effort of a French consortium to provide a unique chemical library for PPI inhibition [196]. A comparative analysis of different PPI-focused libraries was reported in the study by Zhang et al. [197], where they noticed that PPI inhibitors tend to be larger and more hydrophobic than standard drugs and that PPI-focused libraries, although designed using different strategies, tend to share common chemical subspaces. Efforts have also been conducted for the application of combined strategies in the identification of PPI modulators. As an example, Singh et al. [198] proposed a VS protocol using a hybrid SB and LB method, highlighting the benefits of 3D topological descriptors to assess the post-docking output. As a validation set, 11 PPI targets with known active and inactive compounds were considered. In a first step, the compounds were docked using Surflex, and the docked poses were post-processed to calculate the shape similarity and the structural interaction fingerprint similarity to the co-crystallized PPI inhibitor. Notably, the hybrid protocol showed an improved performance for numerous targets, supporting the application of these combined techniques for prospective studies of PPI modulators.

In a different context, it is also worth mentioning here the systemic chemogenomics/QSAR procedure introduced by Cruz-Monteagudo et al. [199], which aimed to generate a disease-relevant pool of ligands by combining phenotypic data with LB and SB information in a sequential process that ended up with a phenotypic VS performed with QSAR models (Figure 9). The first step is the selection of a representative set of ligands targeting a disease with a measurable therapeutic phenotype, such as compounds successfully evaluated in clinical trials. Then, information about the ligand–target interaction, key genes, or protein targets involved in the molecular interactions and reaction networks are compiled to build a disease-relevant chemogenomics space, which should encompass potential targets directly or indirectly (i.e., cascading effects) implicated in the physiological response. Gene ontology is subsequently used to encode the systemic effect of each ligand in fingerprints containing both chemical descriptors and biological information. The codified compounds are split into two classes: ligands that significantly interact with at least one target (phenotype-positive class) and compounds with no significant interaction with any of the targets associated with the desired phenotype (phenotype-negative class). Finally, a QSAR-based VS methodology is performed. This protocol was utilized in a retrospective study aimed at prioritizing ligands acting as neuroprotective agents in Parkinson’s disease, and a significant fraction of the drug candidates used as starting points could be recovered at early fractions of the screened data.

As a final remark, it is worth emphasizing the relevance of curating the activity data in the analysis of the compounds and the preparation of targeted chemical libraries, minimizing the bias introduced by spurious hits, which can arise from a number of unexpected factors, such as covalent modification of specific protein residues or changes in the redox state of a range of ligands. In particular, a large number of cases have been attributed to self-aggregation of the ligand in an aqueous solution [200,201,202]. The tendency of small organic molecules to spontaneously form colloidal self-aggregates can lead to undesired artifacts in the screening of drug-like compounds, resulting in the identification of false positives [203,204]. The colloidal aggregates formed by these types of compounds exhibit common trends, such as the lack of robust structure-activity relationships, as well as the identification of time-dependent noncompetitive-like inhibition [200,205], likely reflecting the nonspecific inhibitory mechanisms related to adsorption of the target protein onto the aggregates or the induction of conformational alterations that affect the protein’s activity [206]. In these cases, a critical parameter to be considered is the critical aggregation concentration of the compound, which turns out to be in the micromolar range for a significant number of aggregating drug-like compounds [202,207]. Overall, this discussion suffices to emphasize the need to perform a detailed curation of the biological data and of the conditions used in experimental assays, as this information may have an unexpected influence on the efficacy of the bioactive compounds.

## 7. Conclusions

In the past decades, VS has been a powerful alternative to high-throughput screening assays due to the reduced expensiveness, the continued progress in computer resources, and the refinement in LB and SB techniques, often leading to hit rate enrichments that outperform the results obtained with experimental screenings. Being the most widely used strategy for disclosing novel hit compounds in the early stages of drug discovery, the success of VS campaigns is also severely affected by the intrinsic shortcomings of both LB and SB methods, which makes it necessary to search for novel computational strategies that exploit the merits of the individual techniques synergistically.

In the last years, we have witnessed a flourishment of different combined LB and SB approaches, ranging from the hierarchical application of techniques in multi-step filtering process to novel methods that integrate LB and SB techniques into a standalone framework. The progress is encouraging, but it can be anticipated that the adoption of these integrated strategies will depend on two main factors. First, an extensive benchmarking of the distinct combination strategies, including a diverse sets of targets covering distinctive structural and physicochemical features, the calibration of different descriptors for similarity measurements, and docking algorithms in retrospective studies should be necessary, eventually complemented with the prospective application to drug discovery projects. These studies should be valuable to judge not only the improvement obtained with the usage of integrated methods relative to either pure LB or SB techniques but, also, to identify the optimal combination strategy in light of the druggability characteristics of the target protein. Second, the ability to implement the combined LB and SB strategies in automated modeling platforms should provide user-friendly access to the screening of targeted-oriented chemical libraries, guidelines for an appropriate design of the combination strategy, and graphical display facilities to analyze the results. The implementation in modern software will be necessary to facilitate the adoption of the combined strategies by the drug discovery community.

## Figures and Tables

**Figure 1 molecules-25-04723-f001:**
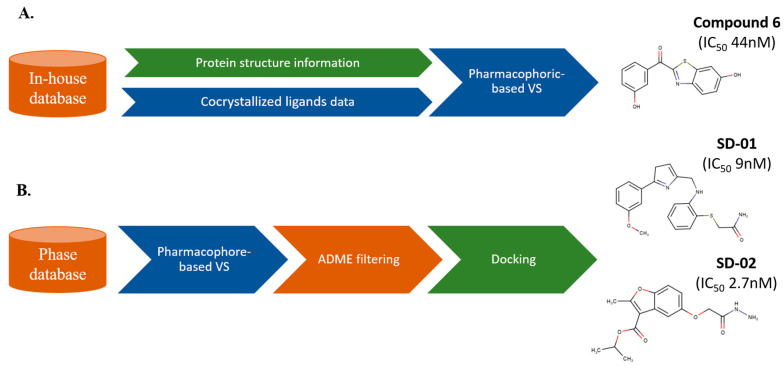
Representative cases of two combined ligand-based (LB) and structure-based (SB) strategies leading to the discovery of potent inhibitors of (**A**) 17β-hydroxysteroid dehydrogenase type 1 (17β-HSD1) and (**B**) histone deacetylase 8 (HDAC8) enzymes. LB and SB methods are highlighted in blue and green, respectively. VS: virtual screening.

**Figure 2 molecules-25-04723-f002:**
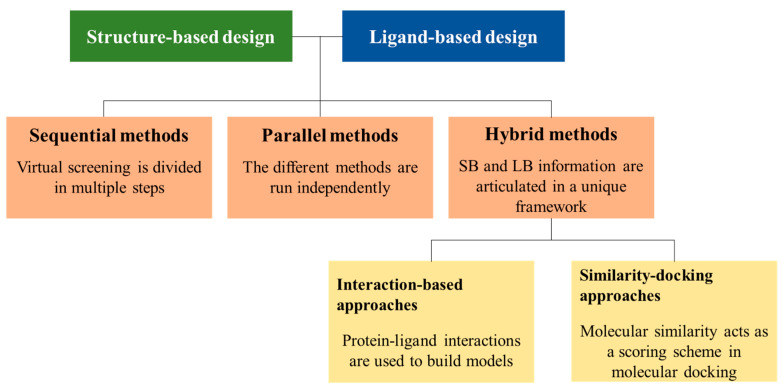
Schematic representation of the three main strategies adopted for combining LB and SB methods.

**Figure 3 molecules-25-04723-f003:**
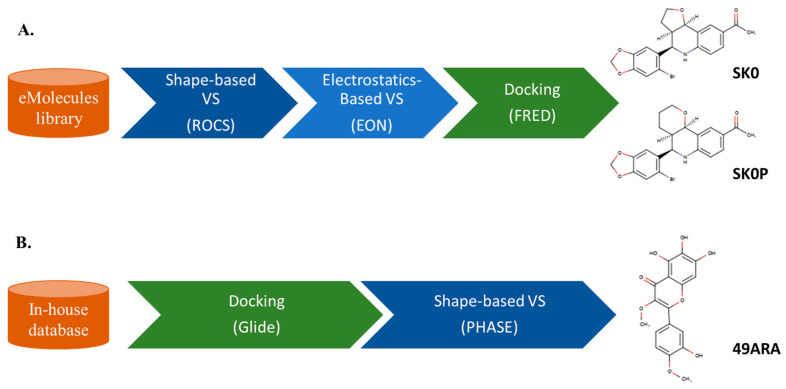
Schematic representation of two sequential VS processes (LB and SB methods are highlighted in blue and green, respectively). (**A**) Sequential application of LB (shape-based and electrostatic-based similarity search) followed by SB (docking). The two most active compounds found, SK0 and SK0P, exhibited antiproliferative activities against SK-BR-3 and MCF7 cell lines in the micromolar range. (**B**) Sequential VS of SB followed by LB over an in-house database. The compound 49ARA was detected among the top-ranked compounds included in the methylene chloride extract of *Artemisia annua* (IC_50_ of 2.2 μg/mL). ROCS: Rapid Overlay of Chemical Structures.

**Figure 4 molecules-25-04723-f004:**
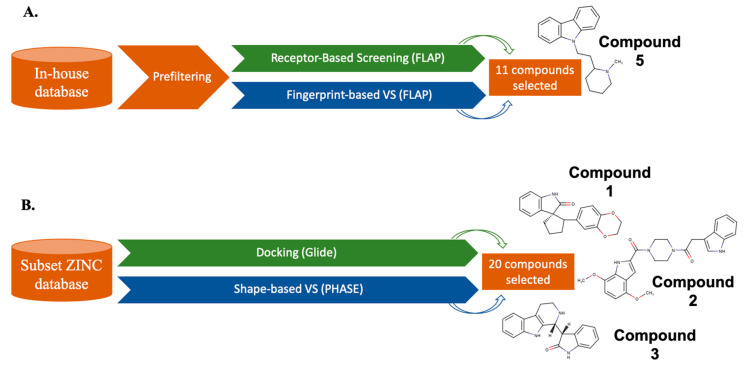
Schematic representation of two parallel VS processes (LB and SB methods are highlighted in blue and green, respectively). (**A**) Parallel application of LB (fingerprint similarity search) followed by SB (pharmacophoric receptor-based screening). FLAP, which allows ligand-ligand and ligand-protein similarity assays, was used for both approaches. The most active compound exhibited an IC_50_ in the micromolar range. (**B**) Parallel VS over a ZINC database subset. Glide was run for SB and PHASE for LB. Among the 20 hits selected, three compounds showed activity in the micromolar range.

**Figure 5 molecules-25-04723-f005:**
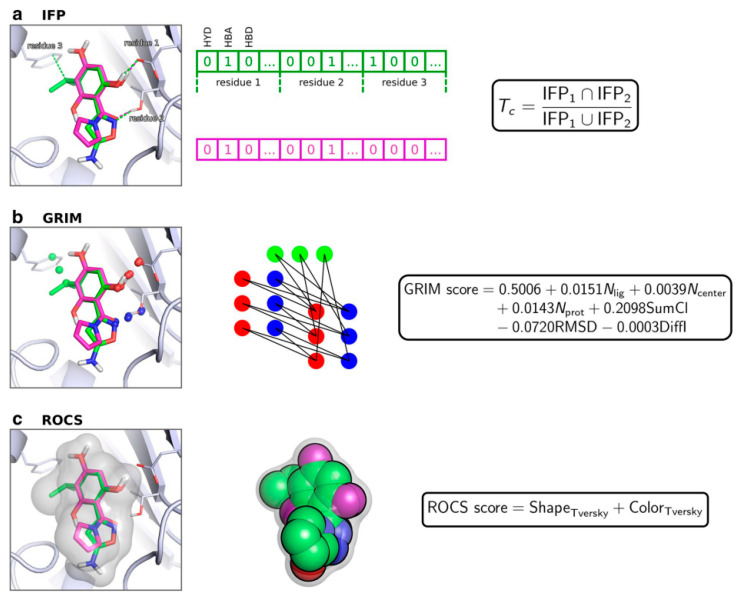
Overview of rescoring methods coupled to interaction fingerprints (IFP), graph matching (GRIM), and ROCS. (**a**) In the IFP score Tc denotes the Tanimoto coefficient. (**b**) In the GRIM score, N_lig_ is the number of aligned ligand points, N_center_ stands for the number of aligned centered points, N_prot_ is the number of aligned protein points, SumCl is the sum of clique weights over all weights, RMSD is the root mean square deviation of the matched cliques, and DiffI stands for the difference between the number of interaction points in the query and the template compound. (**c**) The ROCS score is based on the Tversky coefficient. Reprinted with permission from Springer Nature [79].

**Figure 6 molecules-25-04723-f006:**
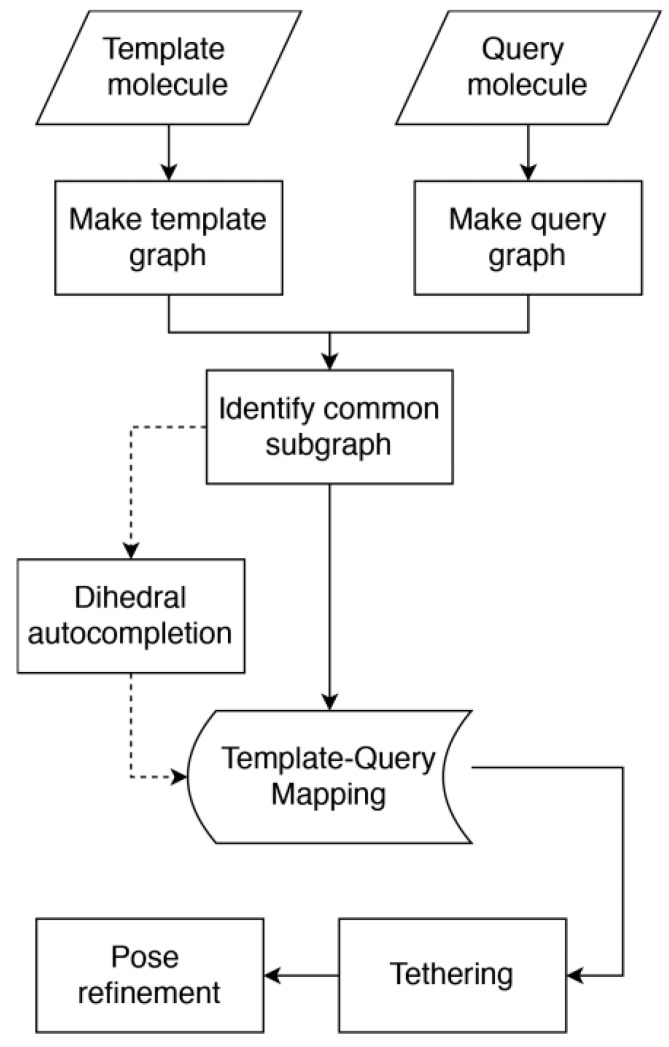
Schematic representation of the flowchart implemented in the SkeleDock algorithm. Reprinted with permission from the American Chemical Society [164].

**Figure 7 molecules-25-04723-f007:**
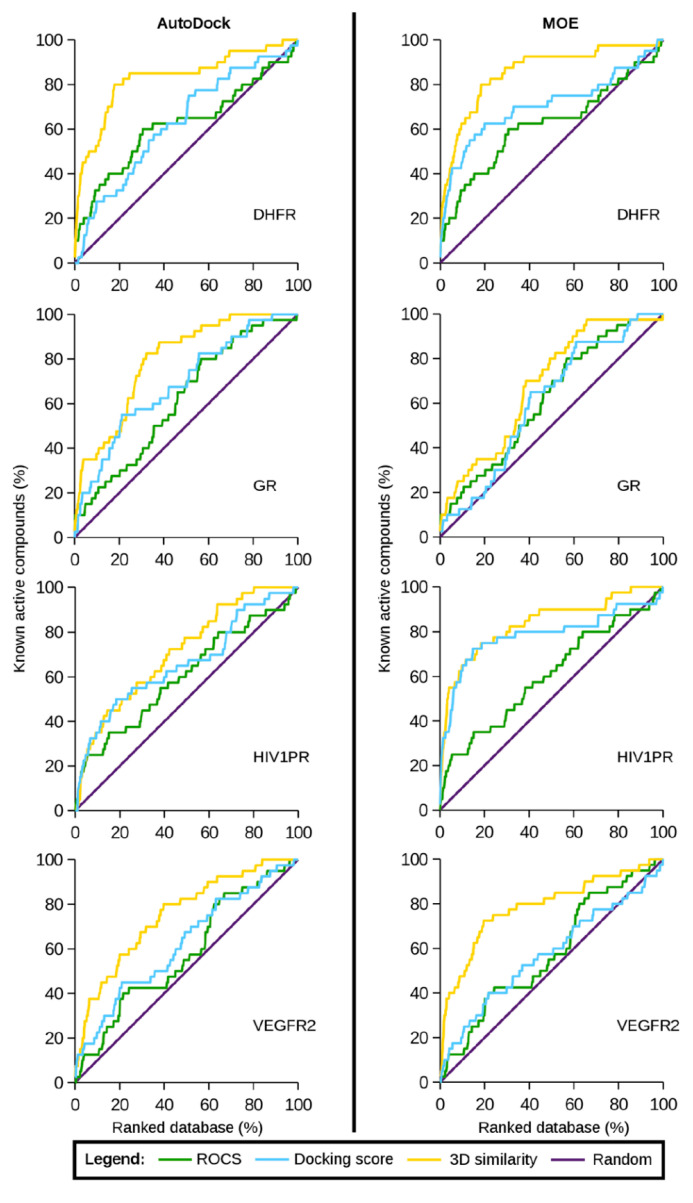
ROC plots obtained for four validation targets (DHFR: dihydrofolate reductase, GR: glucocorticoid receptor, HIV1PR: HIV-1 protease, and VEGFR2: vascular endothelial growth factor receptor-2) using shape-based similarity (ROCS; green), docking (AutoDock [166] and MOE [167]; cyan), and the hybridized method (yellow). Reprinted with permission from the American Chemical Society [76].

**Figure 8 molecules-25-04723-f008:**
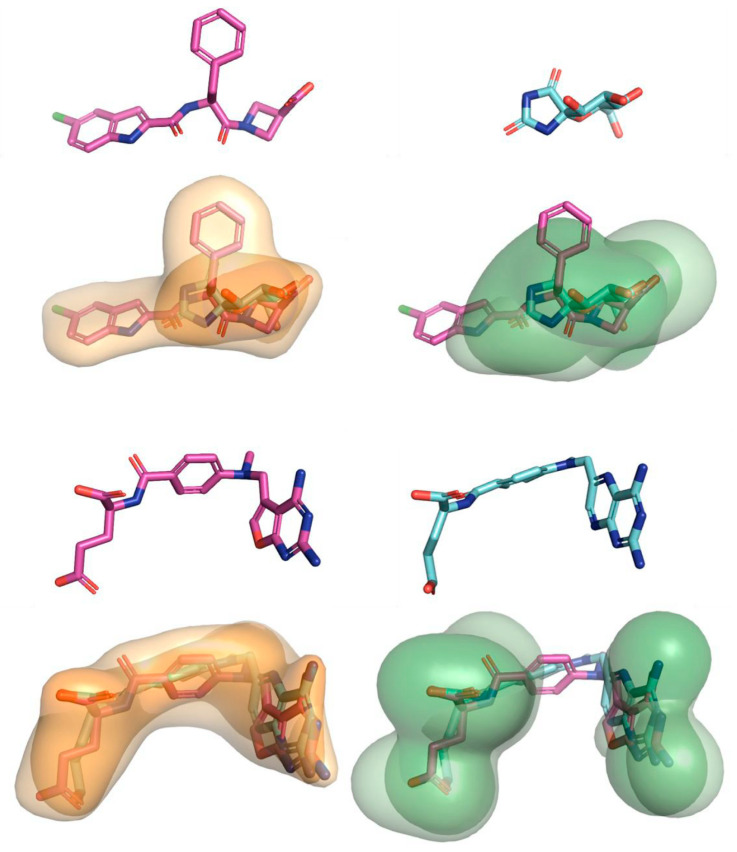
Representation of the hydrophobic molecular overlay exploited by PharmScreen. (Top) Overlay of ZINC02046793 (template) and ZINC1489956 (active) pertaining to glycogen phosphorylase^®^. (Bottom) Molecular alignment of ZINC0384989 (template) and ZINC1529323 (active) pertaining to the dihydrofolate reductase. Orange and green contours denote the fields originated from the cavitation and electrostatic components of the molecular lipophilicity. Reprinted with permission from the American Chemical Society [80].

**Figure 9 molecules-25-04723-f009:**
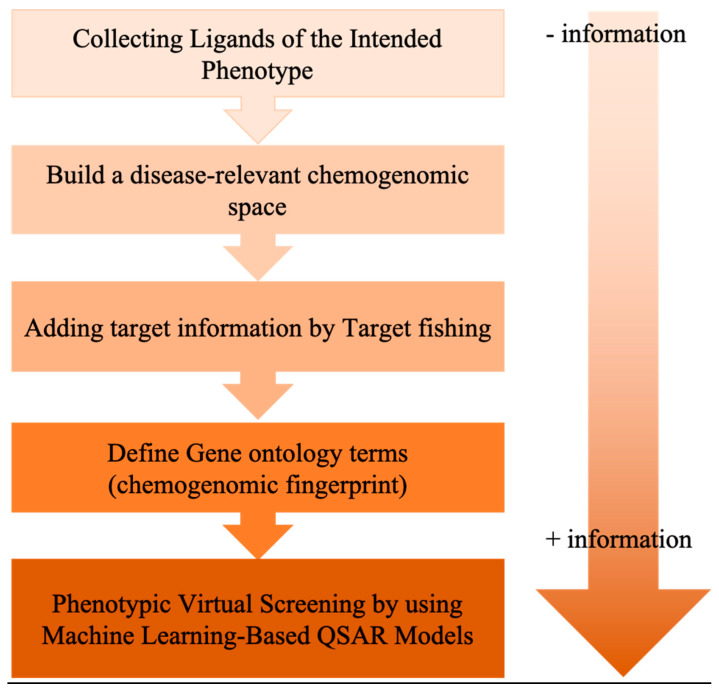
Schematic representation of systemic chemogenomics/quantitative structure-activity relationships (QSARs) for phenotypic VS.

**Table 1 molecules-25-04723-t001:** Description of selected fusion algorithms implemented in parallel ligand-based (LB) and structure-based (SB) strategies. VS: virtual screening.

Algorithm	Description	Case Studies
ADDITION algorithms	Adds together the ranks from the different VS methods rank lists. Standard statistical measures, weighted or not, are used (i.e., sum, average, and median or max. value) to combine rank positions.	[66,67,103,116]
PARETO ranking	Ranks a compound based on how many other compounds are better in all screening methods. Ties could be broken using the sum rank, as example.	[103]
PARALLEL selection	Compounds are alternatively selected among the top-ranked compounds obtained from each screening method until the desired number of compounds is reached.	[81,103]

**Table 2 molecules-25-04723-t002:** Selection of pseudoquery methods categorized by the underlying protein-interaction model.

Model	Methods	Description
Interaction fingerprint-based	SIFt [132]	Fingerprint encoding seven predefined types of target–ligand interactions.
	PLIP [133]	A web service for the detection and visualization of seven protein–ligand interaction types considering a 3D space.
	FLIP [134]	For each residue, 7 different interactions are represented in 10 bits.
	PADIF [135]	Fingerprints with the inclusion of information relative to the strength of interactions and unfavorable ones.
Pharmacophore-based	LigandScout [125]	Pharmacophores derived from six types of nonbonded protein-ligand interactions and volume constraints.
	FLAP [126]	Four-point pharmacophore fingerprints with a shape component.
	IChem [136]	Converts the protein–ligand interaction pattern in fingerprints and graphs.
	TIFP [137]	Encodes a string of unique triplets (two interacting atoms and an interaction pseudo-atom).

**Table 3 molecules-25-04723-t003:** Examples of public and commercial databases (data taken from [177,178]).

Database	Type	No. Cpds
AstraZeneca with Enamine BBs	Proprietary	10^17^
Boehr.-Ing. BICLAIM	Proprietary	5 × 10^11^
CH/PMUNK	Public	>95 × 10^6^
eMolecules Plus	Commercial	5.9 × 10^8^
Enamine Real	Commercial	>300 × 10^6^
EVOspace	Proprietary	1.6 × 10^16^
GDB-17	Public	~166 × 10^9^
Lilly LPC	Proprietary	2 × 10^11^
MASSIV	Proprietary	10^20^
SAVI	Public	~283 × 10^6^
PGVL	Proprietary	3 × 10^12^
PubChem	Public	9.6 × 10^6^
SCUBIDOO	Public	~21 × 10^6^
Sigma Aldrich	Commercial	1.4 × 10^7^
ZINC15	Commercial	2 × 10^6^

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
