# Peer review of "Merging Ligand-Based and Structure-Based Methods in Drug Discovery: An Overview of Combined Virtual Screening Approaches"

_molecules, 2020, doi:10.3390/molecules25204723_

Round 1
Reviewer 1 Report
The manuscript reviews the approaches in which structure- and ligand-based approaches are combined to improve the performances of VS campaigns. The manuscript is clear, well-organized, technically sound and delves into the proposed topic in a complete and meaningful way. The Authors are notable experts in the field who can add their skillful vision to the reviewed papers. Nevertheless and even though the selection of the mentioned papers is clearly subjective, they do not exceed in self-citations but offer a complete picture of the treated topic. On these grounds, the reviewer thinks that the manuscript can deserve publication. The only minor concern involves the inclusion of reference 106 which indeed describes a quite different approach which does not comprise structure-based simulations. This study is surely interesting because it faces the problem about how the phenotypic data can be analyzed by computational approaches. Hence, the reviewer would suggest to insert this paper in a separate section combining this (if it is possible) with other similar computational studies dealing with the same issue.
Author Response
We appreciate the positive remarks made by the reviewer.
The only minor concern involves the inclusion of reference 106 which indeed describes a quite different approach which does not comprise structure-based simulations. This study is surely interesting because it faces the problem about how the phenotypic data can be analyzed by computational approaches. Hence, the reviewer would suggest to insert this paper in a separate section combining this (if it is possible) with other similar computational studies dealing with the same issue.
With regard to this minor remark, we agree with the reviewer's suggestion. We think it is important to highlight this phenotypic strategy, but in the revised version this is discussed in a new section (6. Exploiting chemical libraries and biological data) at the end of the manuscript.
Reviewer 2 Report
The paper presents the list of methods applied for computer-aided drug design. The whole revision is based on recent publications what makes the paper highly actual.
However one important issue is not taken into account.
All mentioned method pay a little attention on the external conditions.
In silico the molecule may work perfectly well however in the real environemnt can not be as good as expected. The main concern is related to the possible self-assembly of the molecules as shown in Fig.3 and Fig. 5. and their possible modifications. The external conditions shall be checked to keep the molecules as individual compounds. The used programs do not take under consideration this particular aspect in a satisfactory form.
I expect any comments on this problem in a Discussion of this paper.
Author Response
We appreciate the favorable comments and evaluation of the manuscript.
The main concern is related to the possible self-assembly of the molecules as shown in Fig.3 and Fig. 5. and their possible modifications. The external conditions shall be checked to keep the molecules as individual compounds. The used programs do not take under consideration this particular aspect in a satisfactory form.
Following the reviewer's recommendation, we have discussed the potential limitations introduced by self-aggregation of the compounds at the end of the manuscript (6. Exploiting chemical libraries and biological data), just before Conclusions. The discussion is accompanied by references 204-211.
Reviewer 3 Report
The authors present a review in the field of virtual screening of small molecule compounds, focusing on combination strategies of ligand-based and structure-based approaches. The manuscript is well-written and the contents are timely. I have a few suggestions, which may make the manuscript more comprehensive.
- Recent efforts in drug discovery focus much on targeting protein-protein interactions (PPI). Therefore, it may be interesting if the authors could add a section about virtual screening specifically focusing on PPI. There should be some differences in strategies between drugs targeting PPI and ones targeting other sites.
- While methods and software for docking and ligand similarity assessments, which are described in this manuscript, are important components in virtual screening, another point we should care about is compound libraries, a starting point of compound screening. Therefore, I was wondering if the authors could write about this topic. For example, what kind of compound libraries are available for free or on a commercial basis, and for what purpose are they used?
- It would be beneficial if the authors could add (selected) references of each algorithm in Table I.
Author Response
We appreciate the reviewer's suggestions, as they have allowed us to discuss specific issues relevant for the topic of this review.
Recent efforts in drug discovery focus much on targeting protein-protein interactions (PPI). Therefore, it may be interesting if the authors could add a section about virtual screening specifically focusing on PPI. There should be some differences in strategies between drugs targeting PPI and ones targeting other sites.
This is a relevant issue given the growing impact of PPI modulators on the design of novel therapeutics. We have briefly discussed this topic in the context of a new section (6. Exploiting chemical libraries and biological data). Specifically, we have presented a selection of diifferent algorithms for the development of PPI-focused chemical libraries, and discussed a recent example of a combined strategy that yielded an improved performance in the identification of PPI compounds. This has been accompanied by new references 193-202.
While methods and software for docking and ligand similarity assessments, which are described in this manuscript, are important components in virtual screening, another point we should care about is compound libraries, a starting point of compound screening. Therefore, I was wondering if the authors could write about this topic. For example, what kind of compound libraries are available for free or on a commercial basis, and for what purpose are they used?
This issue was succintly mentioned in the Introduction in the original version. We agree with the reviewer in the relevance of the chemical library for the success of a VS campaign. Following the reviewers's suggestion, we have expanded this topic, not only presenting several databases for VS (with citation to other reviews, such as 181 and 182), but also expanded the discussion about targeted chemical libraries, primarily with the assistance of Artifical Intelligence techniques. This is presented in section 6 and accompanied by references 176-192.
It would be beneficial if the authors could add (selected) references of each algorithm in Table I.
Some representative examples have been added to Table 1. These cases, although reflecting a personal choice, are intended to be illustrative example for the reader.
Round 2
Reviewer 2 Report
Authors followed my comments.